# Cell Differentiation and Replication during Postnatal Development of the Murine First Molar

**DOI:** 10.3390/biology10080776

**Published:** 2021-08-14

**Authors:** Rudi Balzano, Edoardo Stellini, Carla Mucignat-Caretta

**Affiliations:** 1Department of Molecular Medicine, University of Padova, 35121 Padova, Italy; rudi.balzano@libero.it; 2Department of Neuroscience, University of Padova, 35121 Padova, Italy; edoardo.stellini@unipd.it; 3National Institute of Biostructures and Biosystems, 30161 Rome, Italy

**Keywords:** development, morphogenesis, tooth development, odontoblasts, cell biology

## Abstract

**Simple Summary:**

Teeth are necessary to prepare food for swallowing. The process of teeth development before and after birth may be studied in normal mice and also by reproducing diseases or genetic conditions. However, mice teeth are different from human teeth, since mice have only permanent teeth. Moreover, their incisors continue to grow for the whole lifespan. Hence, it is important to know how the mouse teeth develop. We studied the development of the first molar in mice from birth to weaning and showed that dividing cells are located in a different part of the developing tooth according to age.

**Abstract:**

Various signaling molecular pathways are involved in odontogenesis to promote cellular replication and differentiation. Tooth formation is controlled mainly by epithelial–mesenchymal interactions. The aim of this work was to investigate how cellular replication and differentiation ensue during the formation of the murine first molar in postnatal ages until eruption, focusing on morphogenesis, odontoblast differentiation and cellular replication. Wild-type CD1 mice were examined from birth to weaning. Morphogenesis and interaction between developing epithelial and mesenchymal tissues were evaluated in hematoxylin–eosin and Gomori trichome stained sections. Immunohistochemistry for nestin, which mediates the differentiation of odontoblasts, especially their polarization and elongation, showed that this intermediate filament was apparent already at postnatal day P1 in the apical region of odontoblasts and progressed apically from cusp tips, while it was not present in epithelial tissues. The expression of nuclear antigen Ki-67 highlighted dividing cells in both epithelial and mesenchymal tissues at P1, while one week later they were restricted to the cementoenamel junction, guiding root elongation. The link between odontoblast maturation and cellular replication in the different tooth tissues is essential to understand the development of tooth shape and dimension, to outline mechanisms of tooth morphogenesis and possibly eruption.

## 1. Introduction

Odontogenesis is an example of complex interactions among different tissue components. Understanding its time course may cast light on general processes relevant to development of different organs. Hence, it recently became a major focus for developmental research.

In contrast to humans, who develop deciduous teeth followed by permanent dentition, mice are monophyodont, and their incisors are continuously regenerated throughout life via the activation of the Wnt-beta catenin signaling pathway [1]. The interaction between different tissues is mandatory for correct teeth development as highlighted by maxillary development, which is a pre-requisite for teeth growth [2] and nerve activity, which may influence teeth formation [3]. During early developmental phases, the first branchial arch plays a fundamental role in splanchnocranium shaping, giving rise to maxillary and mandibular processes. Teeth development starts from an epithelial thickening, the dental lamina, which in the mouse originates during embryonal day 12 (E12) [4], giving rise in both the maxilla and mandible to two incisors that grow throughout the mouse’s life and show a remarkable cell heterogeneity [5]. In addition, six molars are present in both the maxilla and mandible, three on each side. The teeth development process goes through a series of stages called bud, cap and bell [6], involving epithelial remodeling through differential growth rate and apoptosis and variations in cell shape driven by cell–cell and cell–matrix interactions.

The literature to date has not explored the interaction between odontoblast differentiation and cellular replication in postnatal morphogenesis of the murine first molar. Little is known about differentiation and replication, whether they are crucial processes in odontogenesis or not and how those two mechanisms guide tooth shaping. 

Given the relevance of mouse wild-type and transgenic models to study teeth ontogenesis, the purpose of this study was to evaluate odontoblast differentiation and cellular replication during the formation of the mouse first molar (M1) in the postnatal period up to weaning, to provide a timeframe for comparison during normal development. We show that nestin expression can be observed throughout all the pre-eruptive stages, peaking immediately before eruption at postnatal day 11. We also present evidence that the proliferation marker Ki-67 is present in all tissues, with the notable exception of enamel knots.

## 2. Materials and Methods

### 2.1. Mice

The procedure was approved by the Ethical Committee of the University of Padova (OPBA, University of Padova, and Italian Ministry of Health, n. 43F3ENEYD) and conformed to the European laws on animal experiments (directive 2010/63/EU). Wild-type CD-1 mice were used. They were kept under standard conditions of light, temperature and humidity, with full access to mouse diet chow and water. Three to five animals of both sexes were used at each age. They were euthanized by cervical dislocation following anesthesia with isoflurane at postnatal (P) days P1, P5, P8, P11, P15 and P19, P19 being the day of weaning, when teeth should possess a fully efficient masticatory function. Newborn mice interact with the environment via the olfactory and tactile sense (whiskers). One week later, the ears open, at P12 eyes start opening, and mice start to leave the nest looking for food independently. Eruption occurs at P15, and at the same time senses are all mature allowing mice to feed freely by themselves.

### 2.2. Histology

After dissection, heads were fixed for 24–48 h in 4% formaldehyde and then washed three times in phosphate-buffered saline (PBS) solution 100 mM, pH 7.2, then dipped in ethylenediaminetetraacetic acid (EDTA) 500 mM, pH 7.4, at room temperature (20 °C) for three weeks. Once decalcification was complete, heads were dissected to obtain hemimandibles, which were then dehydrated through a series of increasing ethanol concentrations, included in paraffin and sectioned at 6 μm in the sagittal plane. Slices were then collected on glass slides.

Paraffin sections were dewaxed in xylene and rehydrated through descending ethanol solutions, followed by deionized water. Two histological staining procedures were used: hematoxylin and eosin staining and Gomori trichrome staining. Dewaxed and rehydrated slides were stained for 10 min in hematoxylin, rinsed and then left for 3 min in eosin. Separately, alternate slides were subjected to Gomori staining first in hematoxylin for 5 min to highlight nuclei then Gomori staining for 10 min. To fix the color, slides were dipped in acidic water (6% acetic acid) for 2 min. Sections were dehydrated and covered with Eukitt (Orsatec, Bobingen, Germany). Samples were then observed using a light microscope.

### 2.3. Immunohistochemistry

Endogenous peroxidase was quenched in deparaffinized sections with 0.3% hydrogen peroxide for 15 min at room temperature in humid chamber. Antigen retrieval was performed with 1 mM sodium citrate, pH 6.0, at 80 °C for 10 min. Slides were then cooled in phosphate-buffered saline (PBS) and permeabilized in Triton X-100 0.5% in PBS for 30 min at room temperature, with 5 min washes in PBS in between. Non-specific binding was saturated with bovine serum albumin (BSA) 1% for 30 min at 37 °C.

A polyclonal antibody anti-nestin (Santa Cruz Biotechnology, Inc.; Dalla, Texas, sc-21247) 1:100 in BSA was incubated overnight at 4 °C in humid chamber, washed in PBS for 5 min and incubated for one hour in the anti-goat secondary antibody conjugated to horseradish peroxidase (Sigma; Mila, Italy AS420) 1:100.

The protein Ki-67 was revealed with a mouse monoclonal antibody (Sigma; P6834) 1:100 in BSA, incubated overnight at 4 °C in humid chamber then washed in PBS for 5 min. The secondary horseradish peroxidase-conjugated anti-mouse antibody (Sigma; Milan, Italy A9044) was incubated for one hour.

Nestin samples were incubated in peroxidase substrate solution (1 mL with 50 μL 1% diaminobenzidine tetrahydrochloride and 50 μL 0.3% H_2_O_2_) for 3 min, washed in deionized water. Ki-67 specimens were incubated in peroxidase substrate solution for 5 min. Alternate sections were counterstained in hematoxylin, then rinsed in flowing tap water, dehydrated in increasing ethanol concentration (50%, 70%, 90% and twice in 100% for 20 s each step) and mounted in Eukitt. Samples were then observed through a light microscope (Leica). Images were acquired using the resident software. Morphometry was conducted following [7]; the percentage of labeled odontoblast area relative to total odontoblast area was calculated for nestin, and the number of labeled cells per square millimeter was calculated for Ki-67. 

## 3. Results

Hematoxylin and eosin staining is used to highlight the dentin in a pinkish color, while ameloblasts and dental follicles appear red because of cells in the active phase of protein synthesis. Gomori trichrome staining in addition highlights collagen fibers with blue pigments.

At P1 (Figure 1A,B), the murine M1 tooth is already in the bell stage with ameloblasts secreting enamel proteins and causing odontoblasts to form a very thin layer of dentin. Odontoblasts are large columnar cells, the cell bodies of which are arranged along the interface between dentin and pulp. They are polarized so that nuclei are aligned away from the newly formed dentin. The pulp chamber is not fully closed. Dentin deposition at P1 starts from the cusp tip (Figure 1A), with the dentin layer becoming thinner in the apical direction. At P5 the chamber is closed, and root formation starts at P11 (Figure 1E,F), suggesting crown full formation. The first vessels can be seen around P11 (Figure 1E,F) within the pulp tissue, but dentin and enamel remains avascular.

Epithelial and follicular tissues are apparent just after birth (P1, Figure 1A,B), where ameloblasts form an organized layer of polarized cells. At P8 (Figure 1C,D and Figure 2A,B) all around the ameloblast layer there is the stellate reticulum, which has a trophic function to ameloblasts, and outer dental epithelium, which is a border to external connective tissues of dental sac. The dental sac later develops the periodontal tissues. 

At P11 (Figure 1E,F and Figure 2C) the formation of the cementoenamel junction (CEJ) can be observed. Apically from it, there is Hertwig’s epithelial sheath (HES), which is involved in determining root number and dimension. As ameloblasts approach the CEJ, they even lose their polarization and tissue organization. Hertwig’s epithelial sheath covers the radicular dentin, and the connective cells pass through it, forming cementum. Subsequently, these epithelial cells form epithelial cell rests of Malassez.

Nestin expression (Figure 3) starts from the tip of cusps and progresses apically to all differentiating odontoblasts. Epithelial tissues do not express nestin. In Figure 3C,D, nestin expression in forming roots indicates that odontoblast maturation proceeds with HES formation. In the apical region there are elongated cells, which apparently do not express nestin; these are probably preodontoblasts ready for ultimate cell cycles before differentiation.

In the mouse, cellular proliferation in early postnatal days is markedly present at cusp peaks (Figure 4A,B). In this region Ki-67 immunolabeling disappears at P5. Epithelial tissues present scarce Ki-67 expression at this age. At older ages concurrently with root formation, dividing cells move towards the apical direction, while no cell in replication is detected within pulp tissue (Figure 5). Intense Ki-67 immunoreactivity can be seen under the CEJ (Figure 4C, arrow). 

At pulp closure, replicating cells are present within the pulp tissue (Figure 5A,D), but as the root forms and HES elongates, the replicating cluster moves apically. Especially in Figure 5B, a groove of Ki 67 positive cells can be seen just under the CEJ, meaning that HES is a sort of signaling center for replication. Hence, HES and the primary enamel knot (PEK) could have a similar instructive function: guiding root elongation and crown shape development, respectively. Noteworthy in Figure 5E is the cervical loop (CL), the exact point where the outer and the inner dental epithelium meet and form HES. Above this point, a modest ribbon-like structure can be seen, that separates root (R) from dental follicle, with a different and more chaotic tissue architecture. At P11, the root has already formed some dentin (De) above odontoblasts. In the apical region some cells expressing Ki-67 are present; this may be compared with Figure 3D, where the same cells do not express nestin. Supposedly, these cells, nestin-negative but in active replication, are preodontoblasts preparing for final steps towards maturation into odontoblasts.

## 4. Discussion

Teeth are made of different cells [6] and tissues: the mineralized ectoderm-derived enamel, the mesenchyme-derived dentin and cementum and the non-mineralized dental pulp, also differentiating from mesenchyme. During development, they interact differently among each other, with the extracellular matrix [8,9,10], with blood vessels, which start differentiating around embryonic day 14, and with nerves, entering pulp at P4 [11] to shape the teeth. These are efficient structures for mastication, defense and voice articulation and encompass an outer portion called the crown, connected through the cementoenamel junction (the neck) to the root, fixed to the alveolar bone. The stages of odontogenesis proceed from epithelial thickening to the dental lamina, tooth bud, cap stage and lastly bell stage, with shaping processes that start early during embryonic development [12]. Enamel is a hard, translucent protective tissue that originates from ameloblasts through a secretory phase, which drives odontoblast differentiation in the mesenchyme, followed by maturation [4,13]. During the bell stage, odontoblasts start to differentiate and secrete pre-dentin followed by dentin, which is the main component of the tooth body in both crown and root. Dentin in mice has a structure reminiscent of bone tissue [14]. Around the dental papilla, Hertwig’s sheath drives cementum formation and root development [15]. The mesenchyme-derived dental pulp hosts pluripotent stem cells. Mineralization progresses through the formation of hydroxyapatite crystals and involves proteins that bind calcium upon phosphorylation by serine-threonine kinases [16]. Hard teeth tissues like cementum and dentin do not undergo remodeling; however, they may be reabsorbed by odontoclasts during development or pathological conditions [17].

Mesenchymal components appear to drive differentiation of epithelial tissues [18], which in turn shape teeth morphology [19] in a compartment relatively isolated from the rest of the body [20]. Five signaling pathways are mainly involved in tooth development: bone morphogenetic factor (BMP), fibroblast growth factor (FGF), sonic hedgehog (SHH), wingless-related pathway (WNT) and ectodysplasin (EDA) [1,12]. In the mouse, around E12 the dental lamina shows an increased proliferative activity in the zone of future teeth, as evidenced by Ki-67 nuclear antigen expression [21]. During the bud stage, epithelial cells penetrate in underlying mesenchyme without any important variation in form and function, and different cadherins are expressed [6]. In the early bud stage, odontogenic potential moves from epithelial to mesenchymal tissues. Progressing through the cap stage, morphological differences between the teeth become patent, as well as the role of cell density in mesenchymal gene expression and nerve growth guidance via semaphorins [1,3]. The dental cap stays in contact with the dental lamina by means of the lateral lamina. During this stage, initial differentiations occur, and epithelial cells form outer dental epithelium (ODE), inner dental epithelium (IDE), stellate reticulum (SR) and enamel knots (EKs) [18]. Mesenchymal cells, on the other hand, form the dental papilla (DP), from which pulp and dentin cells rise, and the dental follicle or sac (DF) which forms periodontal tissues. The enamel organ appears in the cap stage, characterized by enamel knots [6], transitory signaling centers that guide the development of cusp morphology. The dental papilla gives rise to dentin and pulp. During the bell stage, the crown grows, and the root starts to form, while the enamel organ gives rise to Hertwig’s sheath [22]. Dentin and enamel start to form in the latest bell phase, when odontoblasts exit the cell cycle, a process that occurs over six hours [23]. Since odontoblasts become post-mitotic cells, they do not divide any further [23], so positivity to Ki-67 can be detected in epithelial surrounding tissues [21]. In the bell phase, the intermediate filament nestin appears to be mostly expressed [23,24] by differentiated odontoblasts [25]. The differentiation of odontoblasts is cytoskeletal-dependent, and the first cells to differentiate are at the cusp tip. Subsequently, in a wave-like motion, all other cells differentiate in apical direction [26]. When differentiation and mineral deposition is over and the crown has achieved its final morphology, the root starts to form [27]. In the postnatal period, differentiation is completed and teeth erupt, staying firmly anchored to the maxillary bone [28].

At birth, M1 appears to be already at the bell stage, since mineral deposition has already started and ameloblasts and odontoblasts are differentiated to their last stage [18]. In our samples, at P1 enamel and dentin appear already formed, but the dental papilla is still open. The dental papilla closes at P5, defining crown dimension and localization of cementoenamel junction (CEJ). At the CEJ, inner and outer dental epithelium meet and form Hertwig’s sheath, which guides formation and number of roots. 

At this age, tooth pulp innervation is delineating [29], and cusps begin to shape. In a previous work, Lisi et al. [26] described cusp formation and odontoblast differentiation as interdependent processes that are apparently not regulated by the same process. Preodontoblasts still maintain the possibility to divide, and so there should be an epigenetic signal that controls the number of cell cycles before final differentiation in odontoblasts [26]. Odontoblasts are post-mitotic cells that secrete dentin sialophosphoprotein [30] and polarize due to nestin, an intermediate filament found in cells of mesenchymal origins [24]. During the first 10 days of extra-uterine life, root formation also advances [31] with the involvement of epigenetic regulations [32].

The present data show that nestin expression was apparent at pre-eruptive ages, with a major peak at P11. These findings are partially consonant with findings from other laboratories [24,25], suggesting that nestin expression is constantly maintained during the entire bell stage until eruption occurs and nestin immunoreactivity is lost.

Nestin was mostly apparent at the border between the odontoblastic layer and dentin, where odontoblastic processes form, indicating that it is expressed in these processes, which deepen along tubules. Expression of this intermediate filament begins at a cusp peak, and then it moves in the apical direction. Nestin immunoreactivity could also be observed in the forming root.

Until now, no data described the postnatal time course of the proliferation marker Ki-67. We showed that Ki-67 was apparent in all tissues, both mesenchymal and epithelial, except for the enamel knots and odontoblastic layer. This is consistent with the ubiquitous expression of this protein by dividing cells only [33]. Those structures become hollow as the cusps rise. As the crown reaches its final shape, Ki-67 expression moves to the apical region on the bottom of the pulp chamber, where roots develop. 

It is not clear if root elongation happens because of replication of pulp cells or because of differentiation of preodontoblasts into odontoblast. Hertwig’s sheath guides odontoblast differentiation in the root region [15]. Root elongation begins only when the crown is fully formed [27], suggesting that there should be a signaling pathway between the crown and Hertwig’s sheath.

The present data demonstrate presence of nuclear antigen Ki-67, a marker of dividing cells, in all pre-eruptive ages and the time course of its expression in both coronal and root tissues. Furthermore, the intermediate filament nestin is expressed in all pre-eruptive ages in the odontoblast area, showing the progression of differentiation of these peculiar cells. Particularly high levels of nestin were found at P11, at a dawn of eruption.

## 5. Conclusions

The mouse as an experimental model is of utmost importance as a research tool for elucidating the mechanism of odontoblast differentiation and cellular replication in postnatal odontogenesis. Demonstrating the extent of cellular replication and its postnatal developmental pattern could explain tooth shape and, from a broader viewpoint, even defects in shape that could be found in animal models of disease. As an example, nestin guides odontoblast differentiation in both physiological and pathological conditions [34,35]; giving the timeline of its expression may set the milestones of normal development. The present findings are a starting point for better understanding tooth development of mouse M1 molar and can lead to further investigations to outline the mechanisms of cellular maturation and development of the tooth structure, a seminal step in acquisition of oral functions.

Further investigations will also evaluate the connection between the expression of nestin and Ki-67 on one hand, and the crown and root morphogenesis on the other, to highlight the processes that guide M1 eruption through the communication between the crown and Hertwig’s sheath.

## Figures and Tables

**Figure 1 biology-10-00776-f001:**
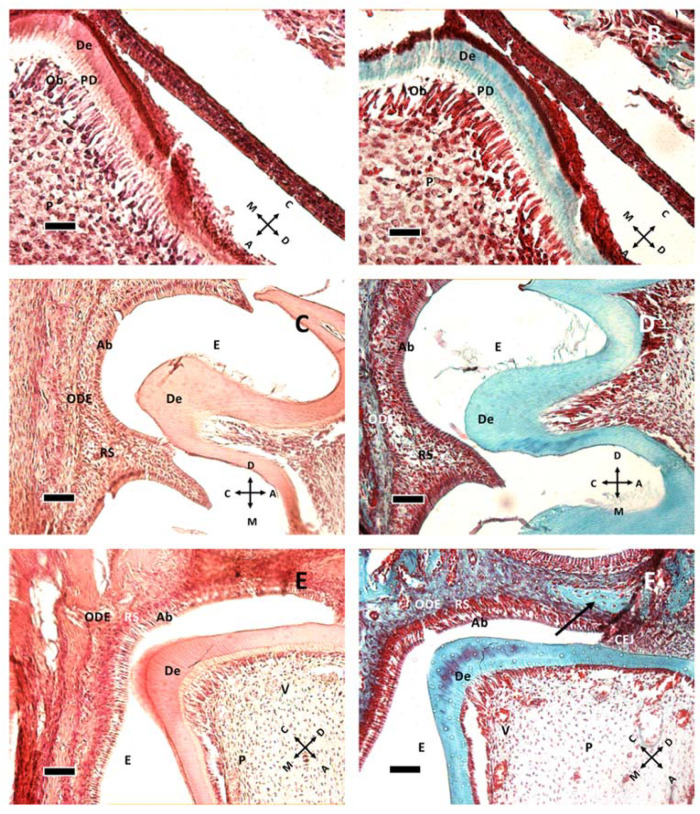
Dentin formation at P1 (**A**,**B**), P8 (**C**,**D**) and P11 (**E**,**F**). The image presents a comparison of dentin in hematoxylin and eosin staining (**A**,**C**,**E**) and Gomori trichrome staining (**B**,**D**,**F**). (**A**,**B**) at P1, the odontoblastic layer, actively forming pre-dentin (PD) and dentin (De), is present. Beneath them, pulp cells (P) can be seen. (**C**,**D**) at P8 the stellate reticulum (RS) lies between ameloblasts (Ab) and the outer dental epithelium (ODE). Enamel (E) cannot be seen due to decalcification procedure prior to staining. (**E**,**F**): At P11, some vessels (V) can be seen in the pulp tissue, and the cementoenamel junction (CEJ) is apparent. The arrow in (**F**) shows a different organization of connective tissue, possibly indicating the difference between dental sac and bone tissue. (**A**,**B**): 40× objective bar = 25 μm. (**C**–**F**) 20× objective, bar = 50 μm. Orientation is given by arrows: coronal (C), apical (A), mesial (M) and distal (D), odontoblasts (Ob).

**Figure 2 biology-10-00776-f002:**
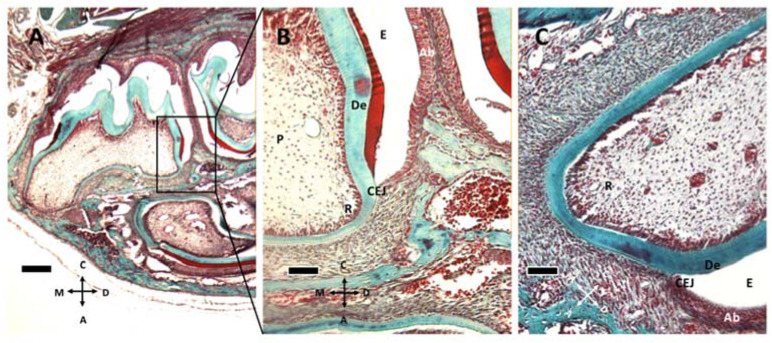
Gomori trichrome staining of M1 at P8 (**A**,**B**) and P11 (**C**). (**A**) M1 and M2 teeth are apparent at P8. (**B**) the box in (**A**) is captured at higher magnification. Enamel (E), ameloblasts (Ab), cementoenamel junction (CEJ) and dentin (De) can be recognized, together with the elongating root (R) and tooth pulp (P). (**C**) At P11, the root tissue appears more structured and well vascularized. (**A**) 5× objective, bar: 200 μm. (**B**,**C**) 20× objective, 50 μm. Orientation is given by arrows: coronal (C), apical (A), mesial (M) and distal (D).

**Figure 3 biology-10-00776-f003:**
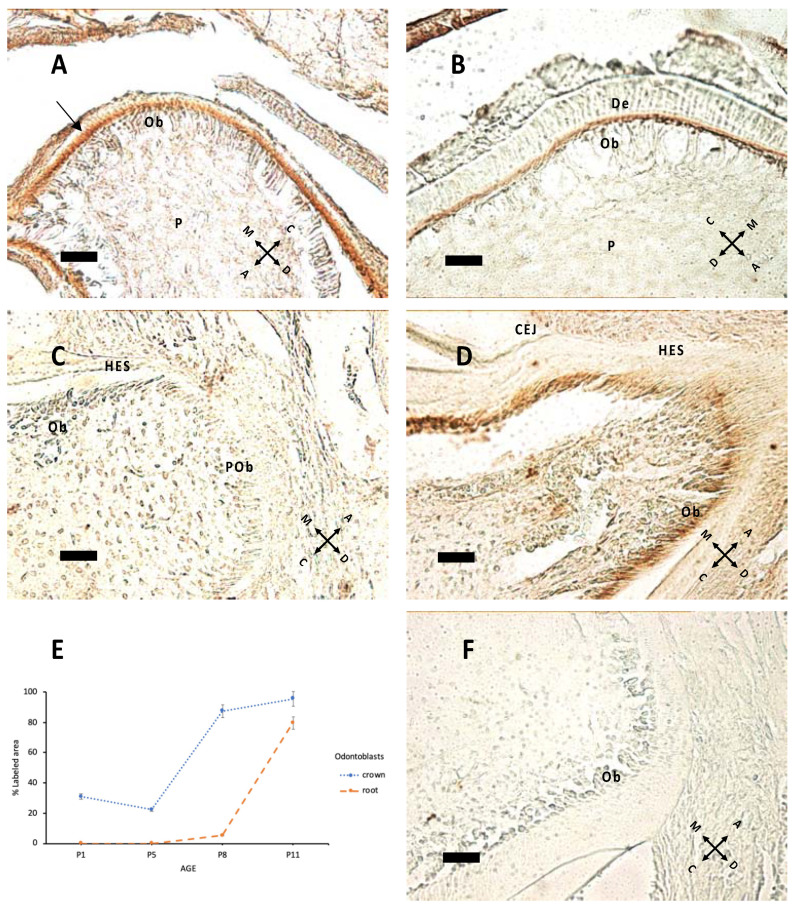
Nestin expression at P1 (**A**), P5 (**B**), P8 (**C**), P11 (**D**). (**A**) The arrow indicates nestin immunolabeling at P1: The signal is higher in the apical region of odontoblasts (Ob) above pulp (P). (**B**) The same pattern of expression is apparent at P5. (**C**,**D**) Root elongation at P8 (**C**) and P11; (**D**) mature Ob in contact with Hertwig’s epithelial sheath (HES). Note the difference in apical region between preodontoblasts (POb), elongated unlabeled cells and subsequent mature Ob in (**D**). CEJ: cementoenamel junction. (**E**) Percentage of labeled area in the odontoblast layer of the crown and root region at different ages. (**F**) Negative control, without the primary antibody, P8 sample. 40× objective, bar = 25 μm. Orientation is given by arrows: coronal (C), apical (A), mesial (M) and distal (D), dentin (De).

**Figure 4 biology-10-00776-f004:**
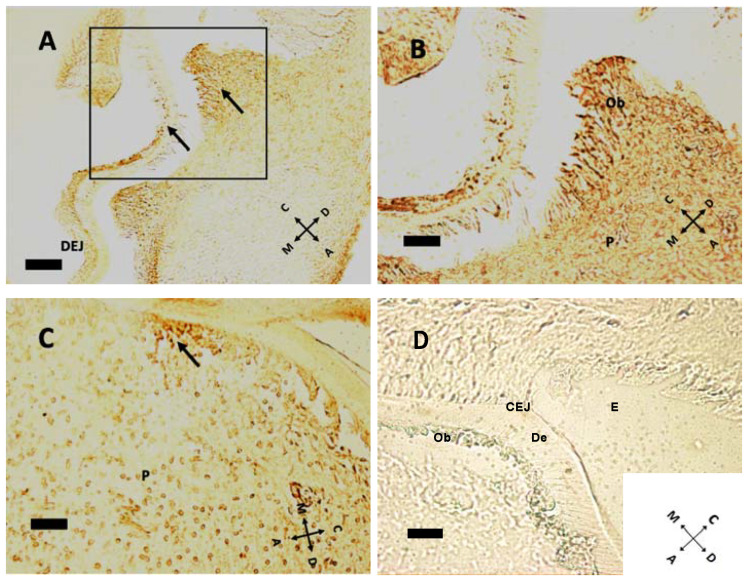
Cell proliferation indicated by Ki-67 immunoreactivity at P1 (**A**,**B**), P8 (**C**). (**A**) At P1, some cellular divisions can be seen at cusp peaks. Arrows indicate proliferating cells (n = 14 and 39, respectively) in both epithelial and mesenchymal tissues. Note the dentin–enamel junction (DEJ). (**B**) Higher magnification of the boxed area in (**A**) Some possibly ultimate cell divisions can be seen in the odontoblast layer, above pulp (P). (**C**) Ki-67-positive cells are all under the cementoenamel junction at P8 (arrow in (**C**)), at a density of 288.25 labeled cells/mm^2^. (**D**) Negative control, without primary antibody, P11 sample. E: enamel; De: dentin; CEJ: cementoenamel junction. (**A**) 20× objective, bar = 50 μm; (**B**–**D**): 40× objective, bar = 25 μm. Orientation is given by arrows: coronal (C), apical (A), mesial (M) and distal (D), odontoblasts (Ob).

**Figure 5 biology-10-00776-f005:**
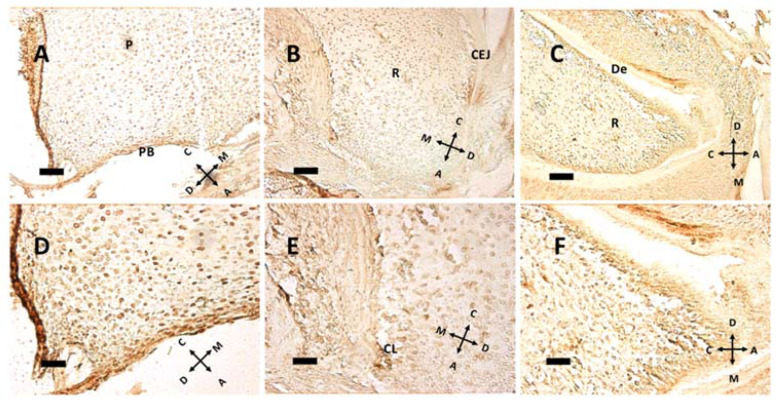
Immunohistochemical detection of the nuclear antigen Ki-67 at P5 (**A**,**D**), P8 (**B**,**E**) and P11 (**C**,**F**). The same fields, captured at lower (**A**–**C**) or higher (**D**–**F**) magnification, are presented. Note the relationship between cell proliferation and root elongation. (**A**,**D**) Some proliferating nuclei can be seen in pulp (P) and the pulpal bottom (PB) at P5, with 263.48 labeled cells/mm^2^. (**B**,**E**) The cervical loop (CL), which guides root (R) elongation, is present, with 226.03 labeled cells/mm^2^. CEJ: cementoenamel junction. (**C**,**F**) While at P8 dentin (De) is not fully formed, it is present as a major deposition at P11, with 382.51 labeled cells/mm^2^. (**A**–**C**) 20× objective, bar = 50 μm. (**D**–**F**) 40× objective, bar = 25 μm. Orientation is given by arrows: coronal (C), apical (A), mesial (M) and distal (D).

## Data Availability

Data are presented within the article. All original slides are available from the authors.

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
