# Peer review of "Cell Differentiation and Replication during Postnatal Development of the Murine First Molar"

_biology, 2021, doi:10.3390/biology10080776_

Round 1

Reviewer 1 Report

it is an interesting paper and well designed. Especially nestin is a very good mark for the purpose of the study.

Author Response

We thank the Reviewer for the positive comments.

Reviewer 2 Report

The authors investigated precise tooth development with histochemical and immunohistochemical analyses, to reveal odontoblasts differentiation and proliferation of the1st molar of mice from birth to weaning. They found that significant cell proliferation detected by Ki-67, a proliferation marker, was observed in various parts in the developing tooth according to age.

I believe that the histological results are technically sound, and the conclusion is clear. In addition, the text is well written.

I would presume, however, this study is highly descriptive, and hence the data in the current manuscript seem to be inadequate to appeal scientific novelty to the readers of the Journal. I would recommend the authors submitting another specific journal to the dental field.

Other minor points to be corrected are listed below.

P1. L13.: “gow” should be "grow"

P2. L44. : “signalling” should be "signaling"

P10. L331.: Delete a period

P10. L347.: Delete a period

Author Response

R2: The authors investigated precise tooth development with histochemical and immunohistochemical analyses, to reveal odontoblasts differentiation and proliferation of the1st molar of mice from birth to weaning. They found that significant cell proliferation detected by Ki-67, a proliferation marker, was observed in various parts in the developing tooth according to age. I believe that the histological results are technically sound, and the conclusion is clear. In addition, the text is well written. I would presume, however, this study is highly descriptive, and hence the data in the current manuscript seem to be inadequate to appeal scientific novelty to the readers of the Journal. I would recommend the authors submitting another specific journal to the dental field.

AUTHORS: We thank the Reviewer for the positive comment. About the Journal choice, we will respect the opinion of the other Reviewers and the final decision of the Editor. Despite being descriptive, we believe the information contained in this manuscript is new and can be of help for persons studying tooth development in mice.

R2: Other minor points to be corrected are listed below.

P1. L13.: “gow” should be "grow"

P2. L44. : “signalling” should be "signaling"

P10. L331.: Delete a period

P10. L347.: Delete a period

AUTHORS: We thank the Reviewer for highlighting the mistakes, all of them have been fixed.

Reviewer 3 Report

The present paper aiming at investigating  how cellular replication and differentiation ensue during the formation of the murine first molar in post-natal ages until eruption focussing on morphogenesis, odontoblasts differentiation and cellular replication. Morphogenesis and interaction between epithelial and mesenchymal forming tissues were evaluated in hematoxylin-eosin and Gomori trichome stained sections. Immunohistochemistry for nestin, which mediates the differentiation of odontoblasts especially their polarization and elongation, showed that this intermediate filament was apparent already at postnatal day P1 in the apical region of odontoblasts and progressed apical-ly from cusp tips, while it was not present in epithelial tissues. 

Overall the manuscript is well written and interesting, but still some minor concerns need to be addressed.

minor concerns:

in the materials and methods section:

line 88 , please correct the word  "thrice"

Immunohystochemical detecion: The quality  of the figures 3, 4 and 5 need to be improved, the figure 4 looks overexposed please revise. Moreover, in the  figure 3, 4 and 5 need to be added the negative ctrls and the morphometric analysis of the positive areas or nuclei positive for Ki-67 and Nestin. 

It is strongly suggest to read the following paper to see how the morphometric analysis has been performed: " In the carotid body, galanin is a signal for neurogenesis in young, and for neurodegeneration in the old and in drug-addicted subjects " published by Marconi et al, and "Selective expression of galanin in neuronal-like cells of the human carotid body" published by Giulio C.D et al.

The manuscript need to be revised by a native english speaker. 

Author Response

R3: The present paper aiming at investigating  how cellular replication and differentiation ensue during the formation of the murine first molar in post-natal ages until eruption focussing on morphogenesis, odontoblasts differentiation and cellular replication. Morphogenesis and interaction between epithelial and mesenchymal forming tissues were evaluated in hematoxylin-eosin and Gomori trichome stained sections. Immunohistochemistry for nestin, which mediates the differentiation of odontoblasts especially their polarization and elongation, showed that this intermediate filament was apparent already at postnatal day P1 in the apical region of odontoblasts and progressed apical-ly from cusp tips, while it was not present in epithelial tissues. Overall the manuscript is well written and interesting, but still some minor concerns need to be addressed. minor concerns:

in the materials and methods section:

line 88 , please correct the word  "thrice"

AUTHORS: this has been modified as requested.

R3: Immunohystochemical detecion: The quality  of the figures 3, 4 and 5 need to be improved, the figure 4 looks overexposed please revise. Moreover, in the  figure 3, 4 and 5 need to be added the negative ctrls and the morphometric analysis of the positive areas or nuclei positive for Ki-67 and Nestin. 

AUTHORS: Figures 3 and 4 have been redrawn with adjusted photomicrographs and three new panels (Fig. 3 E and F, fig. 4 D), with improved resolution. Negative controls with omission of either nestin or Ki-67 primary antibody have been added (New Figures 3F and 4D, respectively).

R3: It is strongly suggest to read the following paper to see how the morphometric analysis has been performed: " In the carotid body,galanin is a signal for neurogenesis in young, and for neurodegeneration in the old and in drug-addicted subjects " published by Marconi et al, and "Selective expression of galanin in neuronal-like cells of the human carotid body" published by Giulio C.D et al.

AUTHORS: The first reference states: “For light microscopy and the data acquisition system, a Leica DM 4000 microscope was used, which was equipped with a Leica DFC 320 digital acquisition system (Leica Cambridge Ltd.; Cambridge, UK). QWin Plus 3.5 software (Leica Cambridge Ltd.; Cambridge, UK) was used to digitize the images and to compute the areas positive for the antibodies.” We tried to adapt the methods to the different labelling of the two antibodies and added the relevant reference, by calculating the labelled area for nestin (see the new Fig. 3E) and the number of labelled cells per square mm for Ki-67. We have added the relevant data for Ki-67 density of cells at the different ages in the text, and a new graph for nestin (new Fig. 3E).

Unfortunately, we were not able to access the second reference (a book chapter) in the short time allowed for revision.

Lastly, we have added two more panels with negative controls (fig. 3F and Fig. 4D).

R3: The manuscript need to be revised by a native english speaker. 

AUTHORS: The manuscript has been revised by a native speaker, all the changes are highlighted in yellow (with the only exception of many articles ‘the’ that have been cancelled).

Round 2

Reviewer 2 Report

I have no more suggestion on this manuscript.